# Robustness of Named Entity Replacements for In-Context Learning

**Saeed Goodarzi**[†]   **Nikhil Kagita**[†]   **Dennis Minn**[†]   **Shufan Wang**[†]

**Roberto Dessì**[∞,π]   **Shubham Toshniwal**[∇]   **Adina Williams**[∞]

**Jack Lanchantin**[*,∞]   **Koustuv Sinha**[*,∞]

[†]University of Massachusetts Amherst; [π]Universitat Pompeu Fabra
[∇]NVIDIA; [∞]FAIR, Meta
sgoodarzitae@umass.edu, koustuvs@meta.com

## Abstract

A key feature of modern large language models (LLMs) is their ability to perform in-context learning, a prompting technique where query-answer demonstrations are shown before the final query. This allows for generalization to novel distributions at inference time where the LLM can learn new rules without parameter updates. However, the choice of demonstrations and their relationship to a particular query can have a profound impact on model accuracy, raising concerns about the true in-context generalization capabilities (Zhao et al., 2021). In this work, we explore the robustness of the in-context learning paradigm by focusing on entities. In particular, we seek to understand the robustness of LLM in-context learning with respect to named entity replacements. We discover a significant variance in downstream performance based on the choice of the named entities, across three popular reasoning tasks and two popular LLMs. Specifically, model accuracy on the test sets can fluctuate between -2.7 to +8.0 points depending on the choice of named entity replacements. Our analysis exposes the sensitivity of LLM in-context learning with respect to named entities, and offers a simple recipe to improve test performance by hyper-parameter tuning the named entities for a given dataset. Code and datasets for reproducing the results are publicly available.[1]

## 1 Introduction

Modern large language models (LLMs) routinely demonstrate in-context learning, where the model can solve a wide range of tasks from a few demonstration examples (or *exemplars*) (Brown et al., 2020). In-context learning, or few-shot prompting, is a powerful learning paradigm as it allows for fast adaptation to new tasks without the need to train

[1]https://github.com/DennisMinn/entities-in-context

[*]Equal Leadership.

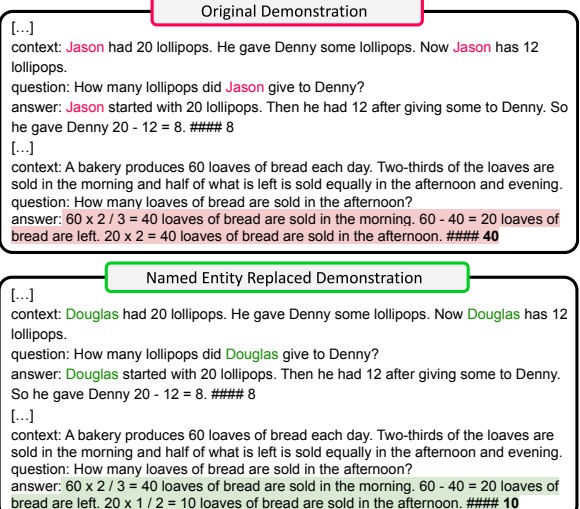

Figure 1: Named entity replacements for In-Context Learning. Example from FLAN-T5 on GSM8k (Cobbe et al., 2021). Here we replace all male entities ($e^o$) in demonstrations with *Douglas*, and observe a change in the query answer prediction (from the incorrect answer *40* to the correct answer *10*). Only 1 out of $k = 5$ demonstrations ($D$) is depicted to conserve space, with the remaining demonstrations denoted by "[...]".

model parameters. However, model performance in the in-context setting can vary significantly with the choice of demonstrations. Previous work has uncovered performance instability when altering the demonstrations, particularly when the demonstrations change semantically (Schick and Schütze, 2020; Zhao et al., 2021; Perez et al., 2021; Gonen et al., 2022a). Naturally, a question arises whether LLMs are robust to *semantically preserving*, minimal alterations to the demonstrations.

In this paper, we seek to address this question using simple, semantically preserving alterations: named entity replacements. Because changing named entities should not, in principle, change the correct response to the query, the model is expected to remain invariant to such modifications. For example, given the demonstration "*Alice has*

*the ball. Alice went to the kitchen. Where is the ball? kitchen.*", the model should be invariant to the choice of the named entity "Alice"—i.e., a model should behave the same if we swap all instances of "Alice" with another named entity, e.g. "Eve". However, we observe that the model behavior is not always robust to such alterations (Figure 1)—prompting us to critically evaluate named entity invariance within in-context learning.

We conduct experiments to investigate the robustness of LLM in-context learning with respect to named entity replacements in three popular named entity heavy reasoning tasks, namely, bAbI, Weston et al. 2015; CLUTRR, Sinha et al. 2019; GSM8k, Cobbe et al. 2021, and two popular models, Flan-T5-XL (Chung et al., 2022a) and GPT-J (Wang and Komatsuzaki, 2021). Our key findings are:

- Neither Flan-T5-XL nor GPT-J are robust to entity replacements for in-context learning. On bAbI tasks, the best replacement entity yields a +8.0 point performance increase (for Task 16), and the worst replacement entity yields a -2.7 point performance drop (Task 12). Similarly, for CLUTRR, we observe a 1.6 point improvement. For GSM8k, we only find <1 point increase/decrease with entity replacements.

- Performance gains from best performing entities are not random. To verify, we ran a bootstrap analysis and observe consistent gains with low variance.

- Best/worst performing entities are task-specific.

- We also analyze frequency, token length and perplexity of replaced entities, and find none of these alone explain our results, introducing an open problem for future research.

## 2 Experimental Setup

The goal of this study is to investigate the robustness of LLMs' in-context learning performance by changing entities in the demonstrations and queries. We consider a fixed set of prompt demonstrations aside from the named entities. By replacing the named entities and observing any accuracy changes, we can better understand the effect of certain entities on a model's accuracy. While many prior works have studied the robustness of LLMs with respect to perturbations (Ribeiro et al., 2020; Jin et al., 2020; Balasubramanian et al., 2020), we focus our analysis on in-context learning. A more in-depth analysis of related work is provided in

Appendix subsection A.1.

### 2.1 In-Context Learning

With in-context learning, a test sample is comprised of a sequence of $k$ demonstrations $D = d_1, d_2, ..., d_k$. Each $d_i$ consists of a context, question and answer string. The final query $q$, consisting only of a context and question string, follows $D$. The goal of in-context learning is to predict the answer to $q$ given a small set of demonstrations at inference time. In other words, learning is done by eliciting activations and not changing model parameters.

### 2.2 Modifying In-Context Learning with Named Entity Replacement

We seek to measure the change in accuracy when we replace named entities in the demonstrations ($D$) and/or query ($q$). We perform entity replacement in the following way.

We first extract the named entities from each demonstration and query using an off-the-shelf NER model, and flag the subject entity, $e^o$, for replacement.[2] We then select a replacement entity $e^r$ from a large set of entities from Tzioumis (2018), which is composed of 4,250 unique first names. Finally, we perform named entity replacement in one of the following three settings[3]: (**D**) replace all $e^o$ entities in the demonstrations, $e^o_{d_1}, e^o_{d_2}, ..., e^o_{d_k}$, (**q**) replace all $e^o$ entities in the query, $e^o_q$, or (**D and q**) replace all $e^o$ entities in both the query and demonstrations, $e^o_{d_1}, e^o_{d_2}, ..., e^o_{d_k}, e^o_q$. Note that each demonstration $d_i$ can contain multiple instances of the entity $e^o$. These settings allow us to isolate and evaluate the effect of named entities in different parts of an in-context learning sample on accuracy.

We run multiple experiments using boostrap sampling (with replacement) on three sets of names, half labeled as 'male' and half as 'female'.[4] For each name $e^r$ in a particular bootstrap set, we replace the original entities $e^o$ with $e^r$ in the test samples according to one of the three previously mentioned settings. We then evaluate the performance on the entity-replaced dataset. This procedure is repeated for each bootstrap set.

---

[2] https://huggingface.co/dslim/bert-base-NER, see Appendix Section A.3 for details on the selection of $e^o$.

[3] Appendix Section A.4 describes each setting in detail.

[4] While our data source provided a binary gender annotation on names based on self-report, we note that this is a proxy and moreover, that using names as a proxy for gender is indicative but can be noisy.

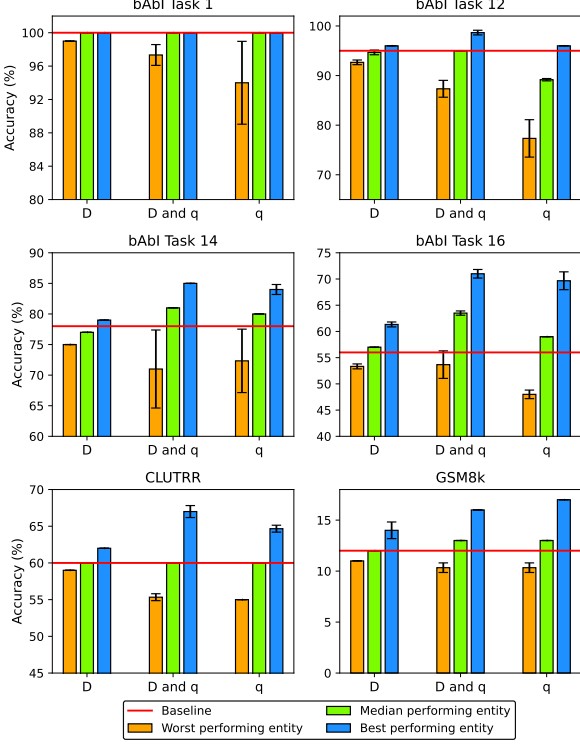

Figure 2: Bootstrap results of Flan-T5-XL on validation sets for 3 distinct sets of 200 names. Error bars represent the variation across bootstrap sets of names.

Baseline results are reported using the original prompt demonstrations and query. However, our replacement strategy introduces a frequency bias, as $e^r$ is used to replace multiple demonstrations in a given input, resulting in artificially increasing the usage of a given entity in an input. Thus, we also report the median accuracy of replacement, which serves as a proxy baseline for replacing all demonstrations and/or query with a random entity.

## 2.3 Models and Datasets

We evaluate our entity replacement strategy on both an encoder-decoder LLM (Flan-T5-XL, 3B; Chung et al. 2022b) as well as a decoder-only LLM (GPT-J, 6B; Wang and Komatsuzaki 2021). We test these models across the following datasets: bAbI (Tasks 1, 12, 14, 16; Weston et al. 2015) , CLUTRR (Sinha et al., 2019), GSM8k (Cobbe et al., 2021). We chose these datasets primarily because they include a significant presence of named entities.

For each dataset,[5] we select five samples from the training set to serve as demonstrations. To maintain grammatical correctness and clarity, we

[5]Except GSM8k, where we select 5 of the 8 prompts from Wei et al. (2022).

only do replacements that preserve gender when a pronoun is contained in the sentence (see Appendix A.5). This step specifically addresses any potential uncontrolled effects on the performance from entity replacement, such as perplexity spikes that may results from swaps where the genders of $e^o$ and $e^r$ differ, and the model is expected to accommodate unexpected pronominal coreference with $e^r$.

## 3   Results

**Performance Across Bootstrap Sets.** We find a noticeable change in accuracy with the best and worst performing entity compared to the baseline or median performing entity. Figure 2 shows Flan-T5-XL results on all validation datasets (analogous GPT-J results are in Figure 5 in the Appendix, and follow a similar pattern). This indicates that specific entities can noticeably alter in-context learning performance. Furthermore, the standard deviations for the median and best performing entity across bootstraps are low, suggesting that the selection of different name sets will likely yield similar median and best performance. However, the variation in the worst performing entities is large.

**Performance Within a Single Bootstrap Set.** To analyze the results in more detail, we measure the performance for Flan-T5-XL and GPT-J on a single bootstrap set of 200 examples. We show the results for bAbI task 14 in Figure 3. When we replace names only in the demonstrations ($D$), the variance in model performance is lower compared to both when we replace names only in the query ($q$) and in both query and demonstration ($D$ and $q$). The ($D$ and $q$) setup exhibits the highest variance, as this scenario involves the maximum change in the tokens of the original prompt. Nevertheless, we see a consistent variance from the baseline by simply replacing named entities in any of the three replacement settings. Other datasets exhibit a similar behavior; see Appendix Figures 6 and 7.

**Best and Worst Entity Generalization.** Thus far, we've used the valid set to demonstrate accuracy variation. To evaluate the best performing entities on the test set, we implement a majority voting method as follows. We first find the 10 best performing entities on the valid set. For each sample in the test set, we take the majority answer prediction from the 10 different best names, denoted "Maj@10". Table 1 shows results of entity replacements across the validation and test sets by replac-

ing entities in the demonstration and query (both $D$ and $q$). We observe a maximum 8.0 point accuracy increase, seen on bAbI task 16.[6] The best and worst performing entities vary by dataset, but overall entities generalize across the validation and test sets, with a uniform increase in accuracy from the best entity. A list of these entities for each dataset and further evaluations are in Appendix Table 2.

For all datasets, there are entities with appreciable deviations from the baseline, and these entities generalize across the validation and test splits. This finding indicates that the observed accuracy shifts are not coincidental and highlights the influence of entity swaps, including those that preserve overall example semantics, such as for names, on model performance.

## 4   Analysis

Why are the models not invariant to entity replacements? We conducted several analyses in attempts to find the root cause of this result, but they were largely inconclusive.

**Does the frequency of names affect how much they shift model performance?** The lack of robustness could well be explained by the frequency effects of named entities in the training corpus of the pre-trained model (Wei et al., 2021). We use the entity frequency data provided by Tzioumis (2018), and assessed performance changes across names with varying frequencies. We do not find any correlation between name frequency and the model performance post entity replacement (see Appendix Section B.3).

**Does the token length of names affect performance?** Another hypothesis of the invariance could be due to the inconsistent tokenization of named entities. Rare names can be tokenized into longer tokens based on the learned subwords of the model. We analyzed token length post tokenization and compared it with model performance, but we are unable to find any meaningful relationship between the two (see Appendix Section B.4).

**Does improved performance result from a decrease in prompt perplexity due to entity replacement?** Finally, the surprisal of the LLM to certain entities could be a promising hypothesis to explain the change in model performance (Gonen et al., 2022b). We compare prompt perplexity af-

---

[6]We did not include bAbI Task 1 because the baseline accuracy is already 100%, leaving no space for improvement.

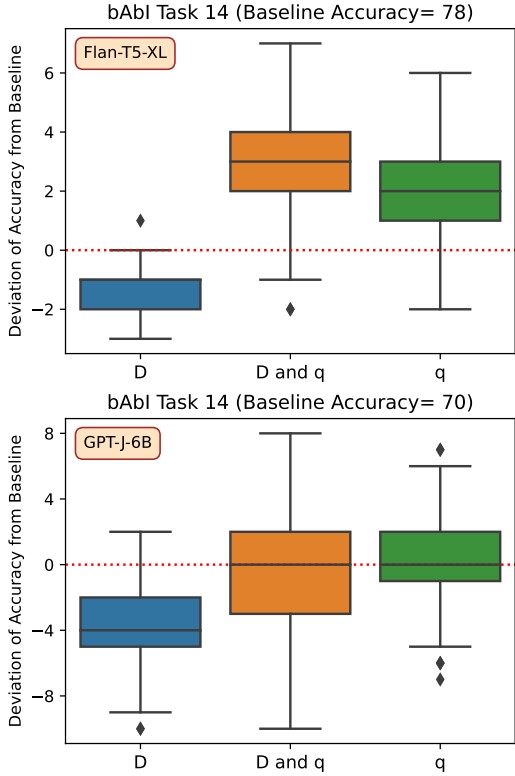

Figure 3: Accuracy for Flan-T5-XL and GPT-J-6B on bAbI task 14 for a single bootstrap set of 200 samples.

ter entity replacement to LLM performance, and yet again we are unable to find a statistical correlation (see Appendix Section B.5). While Gonen et al. (2022b) suggests improvement of model performance with decrease in perplexity, we observe certain entities displaying the opposite behavior.

## 5   Conclusion

In this study, we find that LLMs exhibit a consistent lack of robustness with respect to named entity replacement for the in-context learning setting, fur-

| Dataset | Baseline | | Best Entity | | Worst Entity | |
|---|---|---|---|---|---|---|
| | Valid | Test | Valid | Test (Maj@10) | Valid | Test |
| **bAbI 12** | 95.1 | 95.5 | 98.5 [+3.4] | 99.0 [+3.5] | 92.4 [-2.7] | 93.0 [-2.5] |
| **bAbI 14** | 76.1 | 77.5 | 78.4 [+2.3] | 79.5 [+2.0] | 75.2 [-0.9] | 76.6 [-0.9] |
| **bAbI 16** | 53.5 | 53.3 | 61.7 [+8.2] | 61.3 [+8.0] | 55.5 [+2.0] | 55.9 [+2.6] |
| **GSM8k** | 14.3 | 10.5 | 14.6 [+0.3] | 11.3 [+0.8] | 12.7 [-1.6] | 10.1 [-0.4] |
| **CLUTRR** | 59.0 | 53.0 | 61.8 [+2.8] | 54.6 [+1.6] | 51.4 [-7.6] | 53.1 [+0.1] |

Table 1: Performance of FLAN-T5-XL, with accuracy shift from baseline provided in [square brackets]. Entity replacements are done in the $D$ and $q$ setting. bAbI task 1 reaches perfect accuracy and is thus omitted from this table. Performance between the validation and test datasets is consistent, and we observe a generalizable increase in accuracy across all datasets.

ther questioning the process of reasoning employed by these models (Min et al., 2022). We demonstrate this across three axes: the complexity of the task, the model architecture, and which named entities are to be replaced (in the demonstrations, query, or both). The cause for this effect remains elusive, which opens up exciting research directions. One intriguing result is that the entities that can boost performance are dataset specific. This points to potential avenues for enhancing LLM downstream performance by performing hyperparameter search over a range of entities.

## 6  Limitations

We acknowledge that the selection of names in our study may not provide an unbiased representation of the complete set of possible names. The data provided by Tzioumis (2018) was derived from mortgage applications which may introduce a bias against popular names within marginalized communities. Moreover, this issue is compounded when we sub-sample the set of names due to computational limitations. Additionally, we used a NER model available through Hugging Face to identify $e^o$. It was BERT-based (Devlin et al., 2018) and trained on a CONLL-2003 corpus (Tjong Kim Sang and De Meulder, 2003) (Wolf et al., 2019) , and we anecdotally observed that sometimes it struggles to identify rare names as named entities, for example, it incorrectly identified "TreQuan" as "TreQ." We spot checked for this issue and removed names that posed problems to the best of our abilities.

## 7  Ethics and Broader Impact

In this work, we explore the effect of name swaps in an in-context learning setup and find that particular names can increase accuracy. In our efforts to align names with their respective pronouns, we assumed binary gender, but acknowledge that other genders exist. We made this assumption because our evaluated datasets used binary pronouns, and our names dataset was also only binarily annotated. While our paper assumes name-gender associations, we do not want to discredit negative evidence. A name that is not explicitly associated with a gender doesn't imply evidence of absence of association (imagine that only one person named "Alex" was sampled, and she was a woman—that doesn't mean that there aren't also men or nonbinary people called "Alex"). At the moment, there aren't many annotated name datasets that consists

of gender-agnostic, gender-ambiguous, or gender-neutral names. Hopefully, additional work such as (Dev et al., 2021; Lauscher et al., 2022; Ovalle et al., 2023) will expand such resources, but regardless of these complications, our conclusions are still useful for trying to understand why some names can shift model performance under some settings.

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

# A Appendix

## A.1 Related Work

Several works investigated the robustness of language models with respect to input rewrites (Papernot et al., 2016; Ebrahimi et al., 2017; Gao et al., 2018; Jin et al., 2020). In particular, Balasubramanian et al. (2020) found that while fine-tuned BERT models were brittle in certain tasks, they demonstrated robustness against entity replacement for question answering. Other works have shown that many transformer-based models are insufficiently robust to name swaps (Ribeiro et al. 2020; Agarwal et al. 2020; Lin et al. 2021; Wang et al. 2021, i.a.), Our work differs from these in that we focus on the in-context learning setting.

The ability to do in-context learning first emerged from Brown et al. (2020). Subsequently, many works sought to improve and understand the abilities of LLMs with respect to in-context learning (Liu et al., 2021; Zhao et al., 2021). Perez et al. (2021); Logan IV et al. (2021); Schick and Schütze (2022); Turpin et al. (2023) show an improvement over the original paper in the the "true few-shot learning" setup where there is no access to a large test set to tune in-context learning hyperparameters. Despite the impressive results of this new paradigm, little has been understood regarding the generalization of certain in-context demonstrations.

Gonen et al. (2022a) bring to attention the importance of the specific prompts in model performance. They show that the lower the perplexity of the prompt is, the better the prompt is able to perform the task. Min et al. (2022) show that the actual labels in the prompt are not very important. However, the labels still have a small impact in performance.

In recent years, several studies have explored the significance of prompt quality in few-shot learning, particularly in the context of question-answering tasks. One such study was conducted by Shin et al. (2020) who proposed AutoPrompt, an automated approach for prompt generation in few-shot learning scenarios that has been shown to achieve the highest accuracy. The AutoPrompt model leverages gradient-based prompt search techniques to find the optimal prompt for a given task, thereby reducing the need for human intervention. Additionally, AutoPrompt utilizes Masked Language Models (MLMs) to perform sentiment analysis and natural language inference tasks without the need for fine-tuning, highlighting the potential of MLMs

in few-shot learning scenarios. In another study, Gonen et al. (2022a) introduced a novel technique for prompt generation that involves using a small set of manually created prompts and then applying paraphrasing and backtranslation to generate additional prompts automatically. Specifically, they utilized GPT-3 to perform the paraphrasing step, which involves rephrasing the original prompts while preserving their underlying meaning. The backtranslation step then involves translating the paraphrased prompts into another language and then back into the original language to generate new prompts that are semantically similar to the original prompts but with different phrasing and wording. This approach enables the generation of a large number of diverse prompts without requiring extensive manual effort, which can be particularly useful in few-shot learning scenarios where data is limited. Gonen et al. (2022a) also demonstrated an inverse relationship between the perplexity of the prompts and the model's performance across various tasks.

Our work is related to previous studies as we focus on the true few-shot learning setup for the task of question answering and explore the robustness of NLP models against entity replacement. There are three main ways in which our work differs from previous studies. First, our investigation focuses on analyzing the robustness of language models against entity replacement specifically for the task of question answering in the ICL setup, which has not been explored in previous studies. Second, we utilize left-to-right language models, which are more widely used and powerful than masked language models. Finally, our approach involves prompt-learning instead of fine-tuning, whereby the parameters of the language model are modified to improve its performance on specific tasks. By leveraging the true few-shot learning setup and prompt-learning techniques, we aim to develop a more robust and effective approach for few-shot question-answering tasks that can handle entity replacement and other challenges more effectively.

## A.2 Dataset Details

**bAbI (Weston et al., 2015):** The bAbI dataset consists of 20 tasks that evaluate various aspects of language understanding by NLP models. In our experiments, we select four tasks representing varying degrees of difficulty from the bAbI dataset: task 1, task 12, task 14, and task 16.

```
context: Mary and [e_d1] travelled to the
bathroom. John and [e_d1] travelled to the office.
question: Where is [e_d1]?
answer: office
###
context: [e_d2] and Mary went back to the office.
Daniel and Sandra went to the bedroom.
question: Where is [e_d2]?
answer: bedroom
###
context: Daniel and John moved to the hallway.
[e_q] and Daniel journeyed to the bathroom.
question: Where is [e_q]?
answer:
```

Figure 4: Named entity replacement overview for in context learning. Here we show an in-context learning example with $k=2$ demonstrations before the final query. We consider 3 scenarios: replacing only the demonstration entities $(e_{d1}, e_{d2})$, only the query entity $(e_q)$, or both $(e_{d1}, e_{d2}, e_q)$. All replacements are done with the same replacement entity $e^r$.

**CLUTRR (Sinha et al., 2019):** This dataset is similar to a subset of the bAbI dataset, but contains more difficult multi-step questions. This allows us to determine whether named entity replacement robustness is affected by task difficulty. Given a multi-sentence context describing kinship relations between certain entities and a question, the task is to predict the kinship relation between the entities that can be inferred from the context but is not made explicit. We convert this into a multiple-choice question where we expect the model to pick the answer from the four choices provided.

**GSM8k (Cobbe et al., 2021):** This dataset is a collection of elementary multi-step mathematical reasoning questions. We specifically selected this dataset due to the challenging nature of arithmetic operations, where existing models often struggle to achieve accurate results. We subsampled GSM8k to include only examples which contained identifiable named entities.

The Tzioumis (2018) dataset, contains of 4,250 unique first names, associated with their unigram frequency, and self-reported demographic information.

For each sample in the dataset, which is comprised of a context, question, and answer triplet, we transform it into a formatted string. This transformation involves concatenating the field name, a colon, space, the respective value corresponding to that field, and newline character. We ap-

ply this transformation to $k$ demonstrations: $D = d_1, d_2, ..., d_k$ and the final query $q$, where the answer to $q$ is omitted. We concatenate the formatted $k$ demonstrations and query $q$, delimiting each string with a newline character followed by "###" and a newline character. Figure 4 shows an in-context learning example.

### A.3 Entity Selection

In general, we determine $e^o$ by selecting the first encountered entity within the question, as it often represents the subject of the question. If the question contains interrogative pronouns, such as "who," no entities will be present within the question. In this case, we consider $e^o$ as the first entity encountered within the answer since it typically represents the subject being referred to. If the subject of the question and answer is referred to using pronouns then neither the question nor answer will contain any entities, we assume that noun ambiguity is not an issue and select the first encountered entity within the context as $e^o$.

### A.4 Entity-replacement Scenarios

In the demonstration-only replacement scenario, we perform entity replacement exclusively on the demonstrations $e_d$ while keeping the query unmodified 4. By selectively replacing entities within the context, question, and answer of the demonstrations, we aim to examine the extent to which this replacement strategy influences the model's performance.

In the query-only replacement setting, we solely augment the entities within the query $e_q$ while leaving the demonstrations unaltered Fig 4. This allows us to investigate the impact of enhancing the query's entity representation on the model's ability to generate accurate responses.

Applying both demonstration and query replacement, we replace $e_d$ and $e_q$ in both the demonstrations and query simultaneously. By enriching the entity information in both the context and the question, we explore the combined effect of entity replacement on the overall performance of the model.

### A.5 Ensuring Grammatical Correctness

To begin, we verify the pronouns associated with both $e^o$ and $e^r$ using (gen, 2023). If the gender associated with $e^o$ corresponds with that of $e^r$, we proceed to substitute $e^o$ with $e^r$. However, if there

is no alignment in gender, we refrain from altering $e^o$ and maintain its original form.

# B  Analysis

## B.1  Analyzing Best/Worst performing entities

In order to examine the generalizability of the best and worst performing entities in each dataset, we extracted named entities from the validation dataset and tested them on the test dataset. More specifically, we replaced the named entities in the prompts of the validation dataset with a set of names that included 100 male and 100 female names. We then calculated the accuracy for each name, selecting the 10 names that exhibited the highest performance and the 10 names that showed the lowest performance. Next, we replaced the named entities in the test dataset prompts with these 20 selected names and calculated the corresponding accuracy for the test dataset. Table 2 presents the best and worst performing entities for each dataset along with the absolute percentage of accuracy changes. The ensemble model functions by replacing the named entity in the prompt with the highest performing entity and then generating an answer at each step. After cycling through the 10 best performing entities, it selects the answer with the most votes as the final response. This approach is referred to as majority@10. It's important to note that the bAbI task 1 doesn't have a 'best performing entity' because the baseline accuracy is already 100%, leaving no room for improvement.

Figure 8 provides an example of an generated answer for the GSM8k test dataset. The original prompt failed to produce the correct answer, while the transformed prompt was able to successfully answer the question. The absolute improvement of 0.8% in the accuracy of GSM8k through the majority@10 is remarkable, given the following facts: 1) The baseline accuracy for this challenging dataset is 10.5%, making even a small improvement significant. 2) The number of named entities in GSM8k is fewer than in other datasets, meaning only a few words are replaced in the transformed prompt, as shown in Figure 8. The ratio of replaced words to the total number of words in the prompt is 0.032. 3) Of the 1319 examples in the GSM8k test dataset, 663 contain a male name, 377 contain a female name, and 279 don't contain any named entity in the query. We could have achieved greater changes in percentage if we had only considered examples where the gender of the best perform-

ing entity matched the gender of the name in the query (as replacements wouldn't occur otherwise). However, we chose not to adjust the test dataset to ensure that our accuracy improvement could be compared with other studies.

The consistency in performance between the validation and test datasets in CLUTRR is lower than that in the other datasets. This could be attributed to the structure of CLUTRR, where four options are provided in the prompt, from which the model chooses an answer. This setup introduces a degree of randomness to the generated answer, potentially leading to less consistency between the performance of the validation and test datasets.

## B.2  LLM Architecture

The architecture of LLMs plays a crucial role in determining their behavior and performance, including their vulnerability to entity replacement. The design choices and architectural components of LLMs can influence how they process and generate text, affecting their sensitivity to changes in named input entities. By comparing the Flan-T5-XL, which employs an encoder-decoder architecture, with the GPT-J-6B, which represents a decoder-only architecture, we aimed to investigate the impact of LLM architecture on their brittleness towards entity replacement. The high variance in the performance of GPT-J (as shown in Fig 7) underscores that the brittleness towards entity replacement is not exclusive to the encoder-decoder architecture but is also a feature of decoder-only architecture models.

## B.3  Effect of entity frequency

Frequently recurring named entities in the web, which subsequently appear in the training data, can impact both the model's ppl and performance. Entities appearing frequently in the training data are more likely to be well-represented in the model's understanding, contributing to lower ppl values. This occurs because the frequent presence of these entities reduces the unpredictability of the next word in a sequence, making it easier for the LLM to anticipate. The hypothesis is that names appearing more frequently would contribute to better model performance. To examine this hypothesis, the frequency of names, extracted from (Tzioumis, 2018), were categorized into five groups of equal numbers, labeled G1 to G5, ranging from the most to the least frequent names. Fig 9 presents the performance of Flan-T5-XL across different datasets.

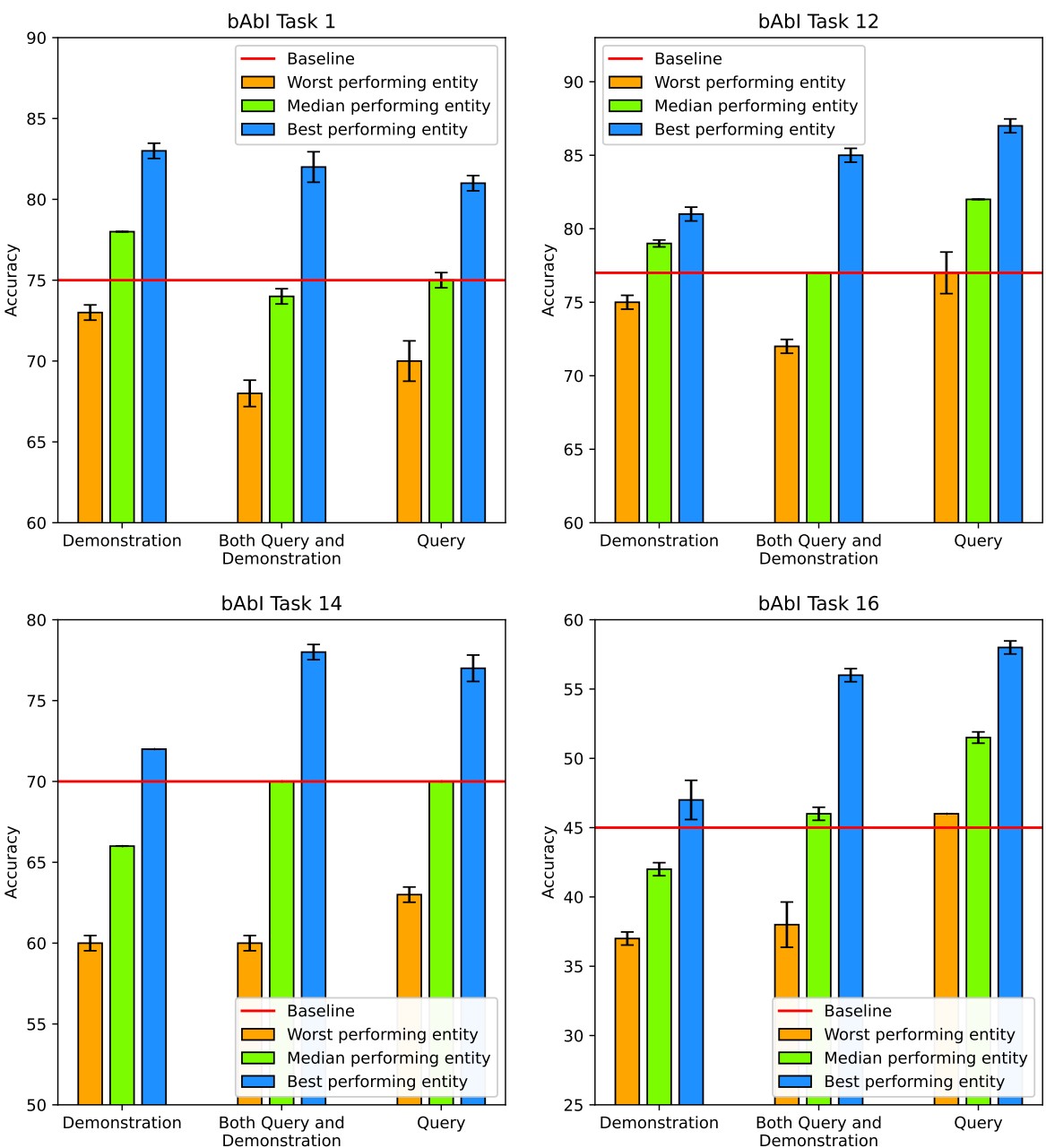

Figure 5: Bootstrap results of GPT-J-6B for 3 distinct sets of names, each comprising 200 names. Error bars represent the variation across different name sets.

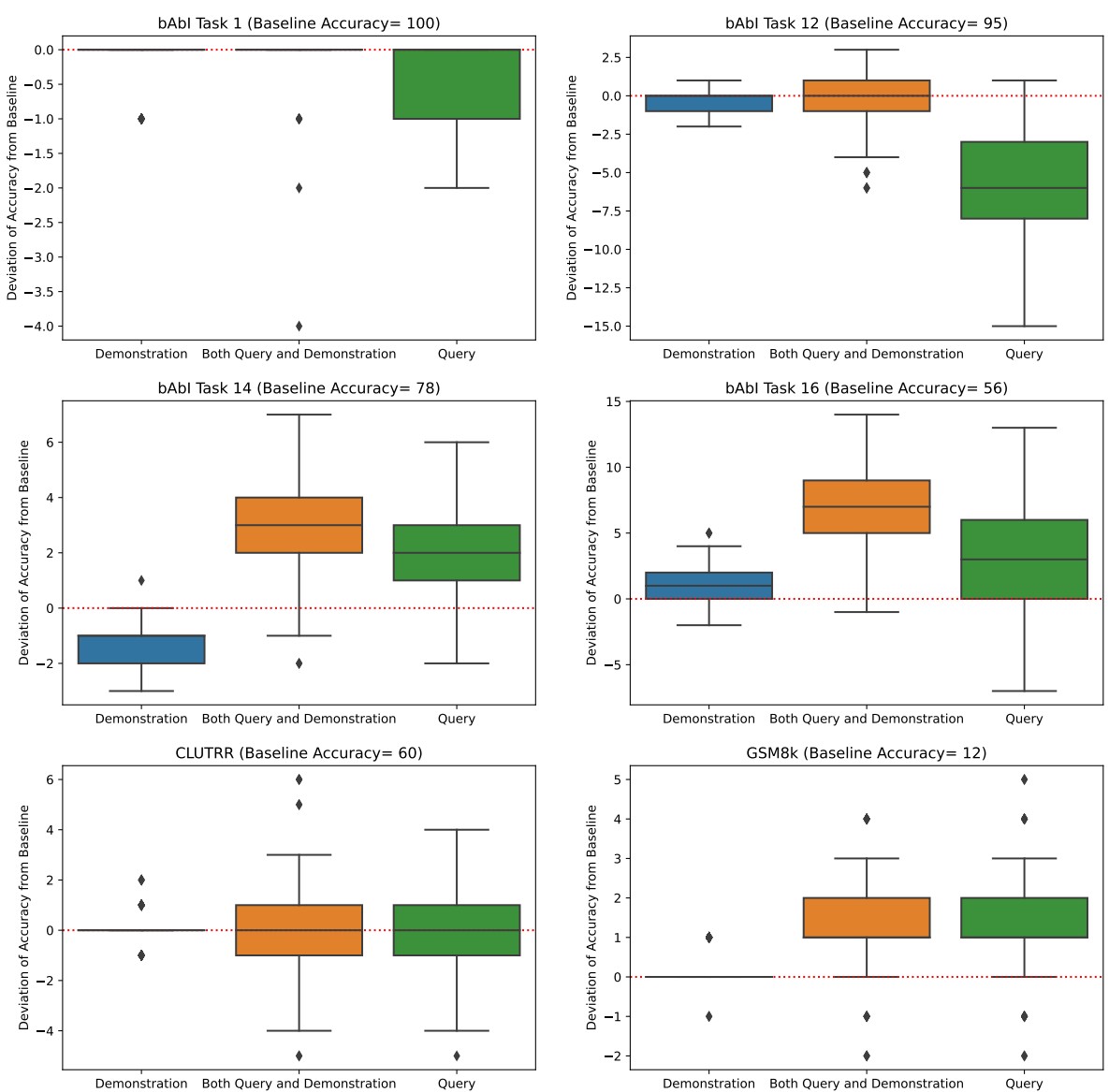

Figure 6: Deviation in accuracy from the baseline (without data augmentation) for Flan-T5-XL across different datasets.

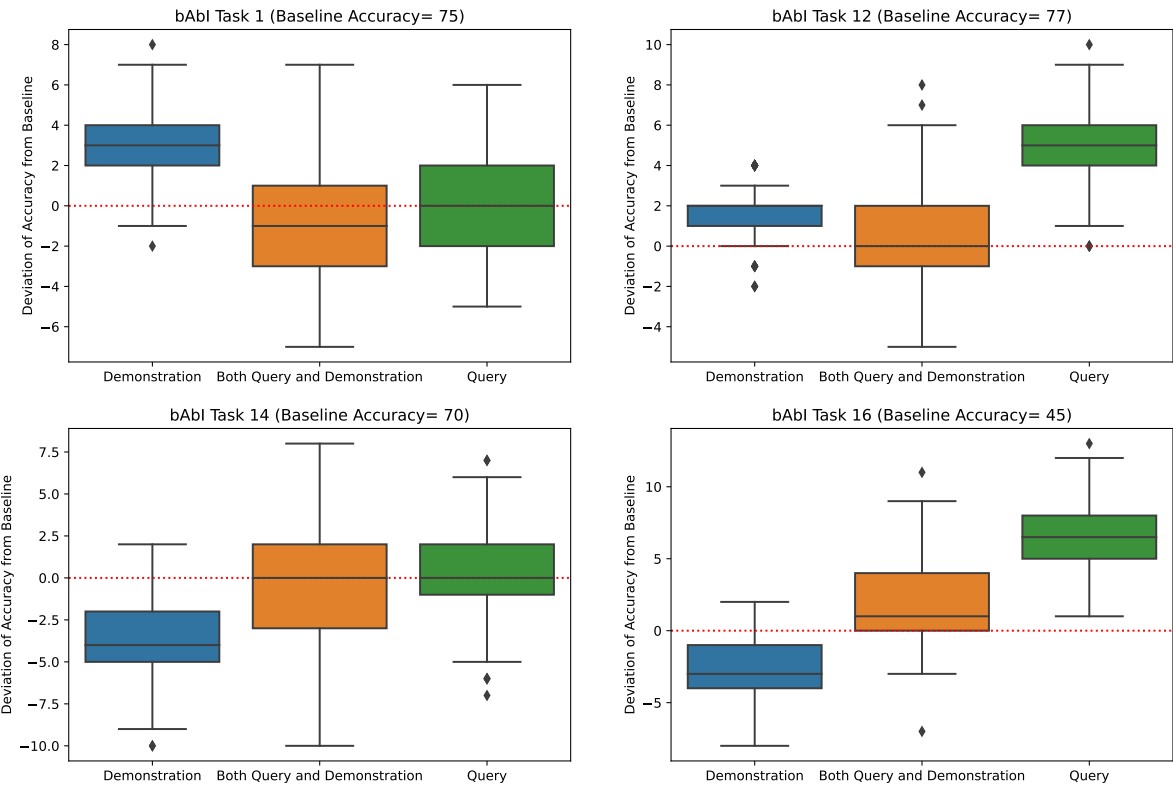

Figure 7: Deviation in accuracy of GPT-J from the baseline across various bAbI tasks. Like the encoder-decoder architecture, the decoder-only architecture also exhibits brittleness towards entity replacement.

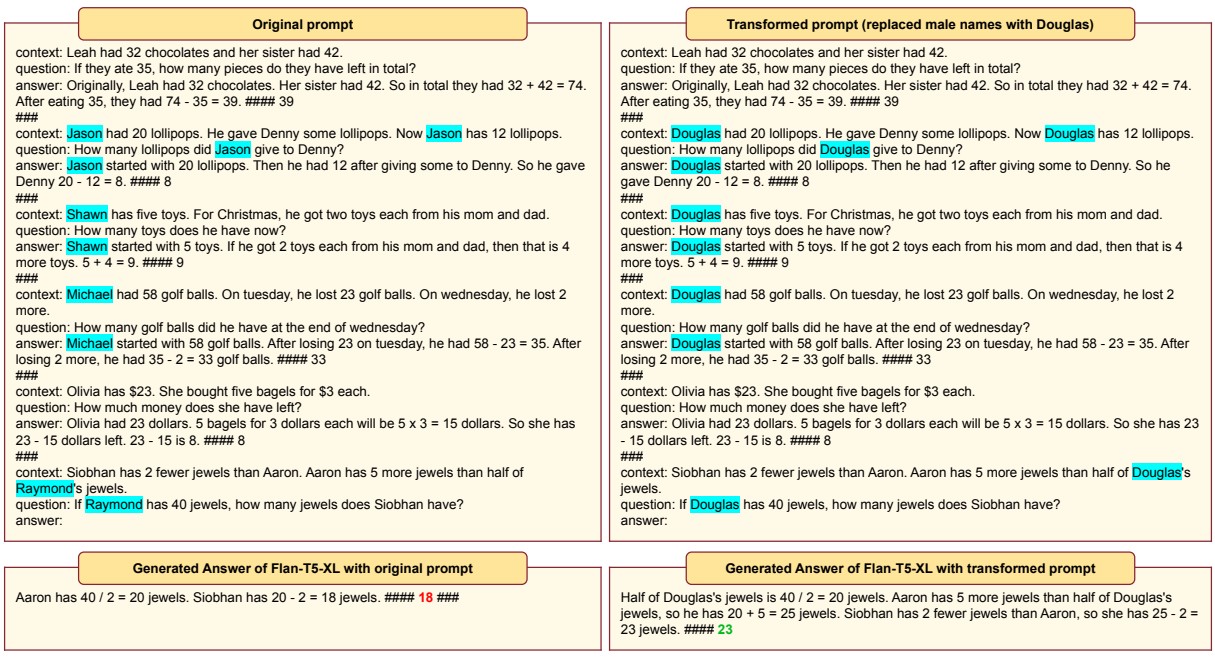

Figure 8: An example of a generated answer from the GSM8k test dataset. In this instance, the original prompt fails to generate the correct answer, whereas the transformed prompt — with the best-performing entity replaced — successfully produces the correct response. This generated text is reproducible using a greedy decoding.

| Dataset | Index | Best performing entity | | | Worst performing entity | | |
|---|---|---|---|---|---|---|---|
| | | entity | Acc change (valid) | Acc change (test) | entity | Acc change (valid) | Acc change (test) |
| **bAbI task 12**
**valid baseline Acc = 95.1**
**test baseline Acc = 95.5** | 1 | Penelope | 3.6 | 3.0 | Mihai | -4.0 | -2.3 |
| | 2 | Loreen | 3.6 | 3.0 | Raisa | -4.0 | -3.6 |
| | 3 | Dorcas | 3.6 | 3.2 | Ethel | -3.7 | -3.5 |
| | 4 | Toby | 3.4 | 3.4 | Migdalia | -3.0 | -4.0 |
| | 5 | Flora | 3.3 | 2.9 | Azucena | -2.6 | -3.2 |
| | 6 | Nataliya | 3.3 | 3.4 | Bhavna | -2.6 | -3.6 |
| | 7 | Nathaniel | 3.3 | 3.4 | Surinder | -2.1 | -0.9 |
| | 8 | Lucretia | 3.3 | 3.3 | Sanjeev | -1.9 | -1.7 |
| | 9 | Keiko | 3.2 | 3.0 | Armida | -1.8 | -1.1 |
| | 10 | Bernardo | 3.2 | 3.1 | Jianwei | -1.7 | -1.4 |
| | **Majority@10** | | | **3.5** | | | |
| **bAbI task 14**
**valid baseline Acc = 76.1**
**test baseline Acc = 77.5** | 1 | Erich | 2.9 | 2.0 | Frederick | -1.4 | -1.2 |
| | 2 | Jacqueline | 2.6 | 1.8 | Raisa | -1.3 | -1.3 |
| | 3 | Zoltan | 2.4 | 2.3 | Maryellen | -0.5 | -0.6 |
| | 4 | Chrystal | 2.3 | 1.5 | Meera | -0.5 | -0.5 |
| | 5 | Petra | 2.2 | 1.1 | — | — | — |
| | 6 | Pravin | 2.2 | 1.8 | — | — | — |
| | 7 | Claudio | 2.2 | 1.0 | — | — | — |
| | 8 | Sanjeev | 2.2 | 1.1 | — | — | — |
| | 9 | Soledad | 2.1 | 2.1 | — | — | — |
| | 10 | Prashant | 2.1 | 2.3 | — | — | — |
| | **Majority@10** | | | **2.0** | | | |
| **bAbI task 16**
**valid baseline Acc = 53.5**
**test baseline Acc = 53.3** | 1 | Loyd | 8.9 | 8.4 | — | — | — |
| | 2 | Armida | 8.7 | 8.1 | — | — | — |
| | 3 | Brannon | 8.7 | 9.5 | — | — | — |
| | 4 | Pravin | 8.4 | 8.8 | — | — | — |
| | 5 | Bozena | 8.1 | 8.2 | — | — | — |
| | 6 | Surinder | 8.0 | 7.2 | — | — | — |
| | 7 | Bonita | 8.0 | 7.1 | — | — | — |
| | 8 | Trena | 8.0 | 6.9 | — | — | — |
| | 9 | Rosalia | 7.9 | 6.2 | — | — | — |
| | 10 | Leopoldo | 7.7 | 9.3 | — | — | — |
| | **Majority@10** | | | **8.0** | | | |
| **GSM8k**
**valid baseline Acc = 14.3**
**test baseline Acc = 10.5** | 1 | Douglas | 0.5 | 0.9 | Leopoldo | -2.5 | -0.8 |
| | 2 | Juliann | 0.4 | 0.2 | Youssef | -1.9 | -0.3 |
| | 3 | Kathleen | 0.3 | 0.4 | Epifanio | -1.7 | 0.5 |
| | 4 | Ernst | 0.3 | 0.4 | Sanjeev | -1.6 | -0.8 |
| | 5 | Tonja | 0.3 | 0.0 | Zoltan | -1.5 | -0.3 |
| | 6 | Angela | 0.2 | 0.2 | Jianwei | -1.4 | -0.4 |
| | 7 | Hina | 0.2 | -0.3 | Abelardo | -1.3 | -0.7 |
| | 8 | Diego | 0.2 | 0.4 | Alphonse | -1.3 | -0.7 |
| | 9 | Bradley | 0.1 | 0.9 | Dagoberto | -1.3 | -0.5 |
| | 10 | Jonathan | 0.1 | 0.8 | Nathaniel | -1.2 | -0.1 |
| | **Majority@10** | | | **0.8** | | | |
| **CLUTRR**
**valid baseline Acc = 59.0**
**test baseline Acc = 53.0** | 1 | Ashlee | 4.4 | 1.2 | Rob | -8.2 | 0.6 |
| | 2 | Penelope | 3.4 | 3.4 | Ahmad | -8.0 | -0.2 |
| | 3 | Allyson | 3.2 | 1.0 | Mihai | -7.6 | 0.6 |
| | 4 | Chrystal | 2.8 | 2.6 | Frederick | -7.6 | -0.4 |
| | 5 | Keiko | 2.8 | -4.4 | Martin | -7.4 | 0.6 |
| | 6 | Alisha | 2.6 | 3.0 | Jake | -7.4 | -0.4 |
| | 7 | Rosalba | 2.4 | 1.8 | Edward | -7.4 | 0.2 |
| | 8 | Jacqueline | 2.2 | 2.0 | Erik | -7.4 | 0.2 |
| | 9 | Zoltan | 2.2 | 1.2 | Giovanni | -7.4 | -0.4 |
| | 10 | Dillon | 2.0 | 1.4 | Douglas | -7.2 | -0.2 |
| | **Majority@10** | | | **1.6** | | | |

Table 2: Performance of the best and worst performing entities in the validation and test datasets. The green color signifies consistent performance between the validation and test datasets, while the orange color indicates inconsistency. The entities, selected based on their performance in the validation dataset, show consistent behavior in the test dataset, suggesting that they generalize in each dataset. We have noted less than 10 entities in some columns because the remaining entities did not reduce the accuracy compared to the baseline.

Contrary to our hypothesis and expectations, there appears to be no correlation between the frequency of names and the performance of LLMs in entity replacement.

### B.4 Effect of entity token length

The token length of a prompt can directly influence the performance of a QA task, particularly as it relates to the computational demands on the model. A longer prompt, represented by a larger number of tokens, increases the amount of data the model must process, potentially straining the model's capabilities and affecting the overall efficiency and accuracy of the task. Changes in the token length of a replaced name, which in turn modify the prompt's overall token length due to multiple instances of the name within the prompt, provide an interesting variable for analysis. We delved into this potential effect by examining the model's performance in relation to the token length of names, as illustrated in Fig 10. Contrary to expectations, our study found no observable correlation between the model's performance and the token length of names. This suggests that, within the ranges examined, the token length of names does not significantly affect the model's performance in QA tasks.

### B.5 Perplexity Analysis

Our expectation was to observe a strong inverse correlation between the perplexity and accuracy of the model across all datasets, according to the findings reported in (Gonen et al., 2022a). However, we noted this inverse correlation only in some datasets, such as the bAbI task 14 (as shown in Figure Fig 11), but not in all of them. A plausible explanation for this inconsistency could be significant differences in our methodology compared to the approach adopted in (Gonen et al., 2022a). In our study, we limited our alterations to substituting named entities within the prompts, as opposed to changing the entire prompt. Therefore, this discrepancy in data augmentation techniques could potentially account for the lack of a significant correlation in some datasets.

Fig 12 shows the deviation of ppl from the baseline across various tasks. Regardless of the task, a more reduction in prompt ppl was observed when entities were replaced both in demonstrations and queries, compared to the other two data augmentation settings. This can likely be attributed to the repeated usage of the same replaced named entity within the prompt. The frequent repetition

of the same named entity can contribute to lower ppl due to the decreased uncertainty and increased predictability of the next word in the sequence.

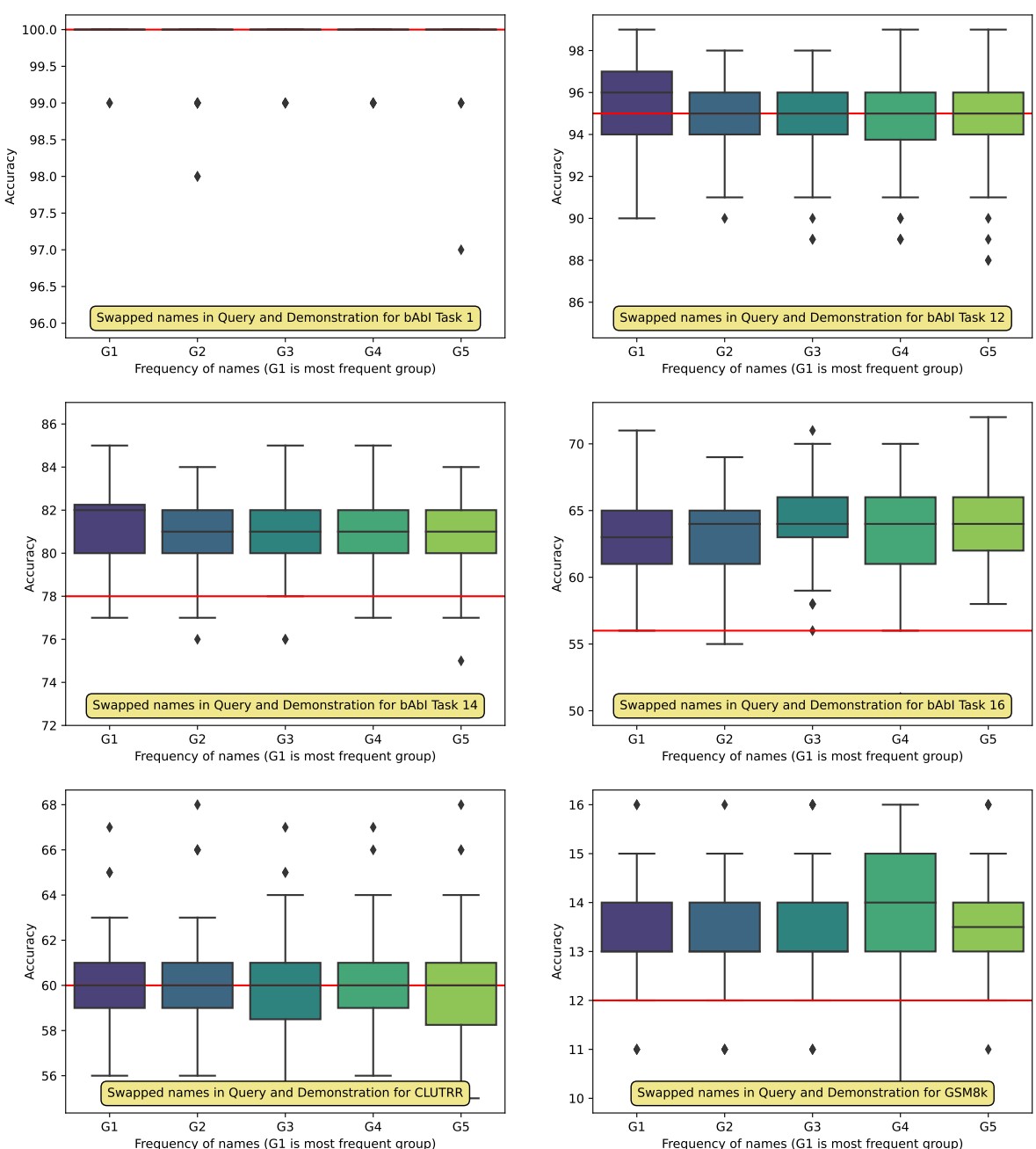

Figure 9: The performance of Flan-T5-XL in relation to the frequency of names derived from a demographic dataset. Contrary to expectations, no correlation appears to exist between the frequency of names and the performance of Flan-T5-XL.

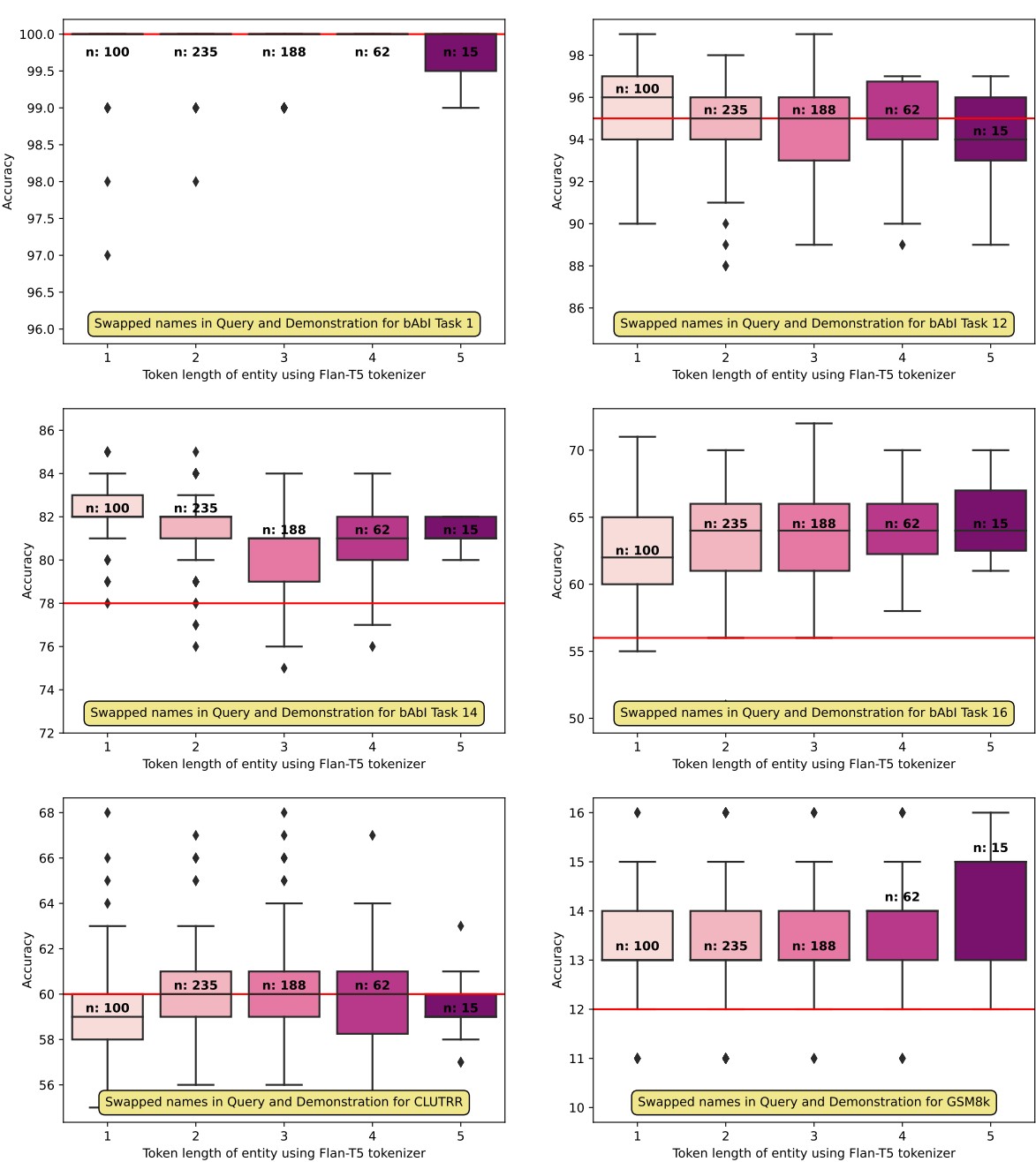

Figure 10: The performance of various names in relation to their token length when using Flan-T5-XL. There is no apparent correlation between the performance and the token length in names. The quantity of names corresponding to each group of token lengths is illustrated on the plot.

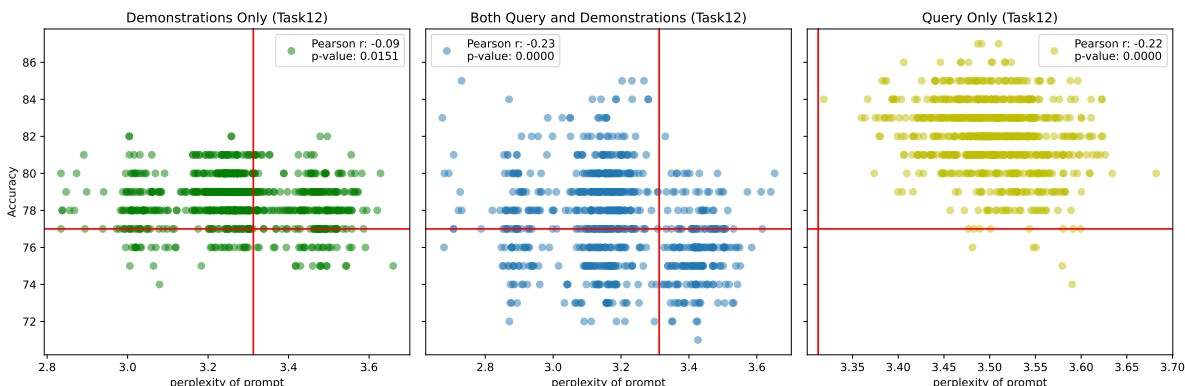

Figure 11: Correlation between prompt perplexity (ppl) and LLM performance for task 14 and GPT-J-6B across various data augmentation settings. The accuracy and perplexity of prompt for the baseline are shown by red lines. Although a statistically significant inverse correlation between accuracy and ppl is observed in this dataset, this trend is not consistent across all datasets. The absence of this trend, contrary to expectation, could be attributed to the nature of our data augmentation approach, which only involved entity replacement instead of entire prompt modification.

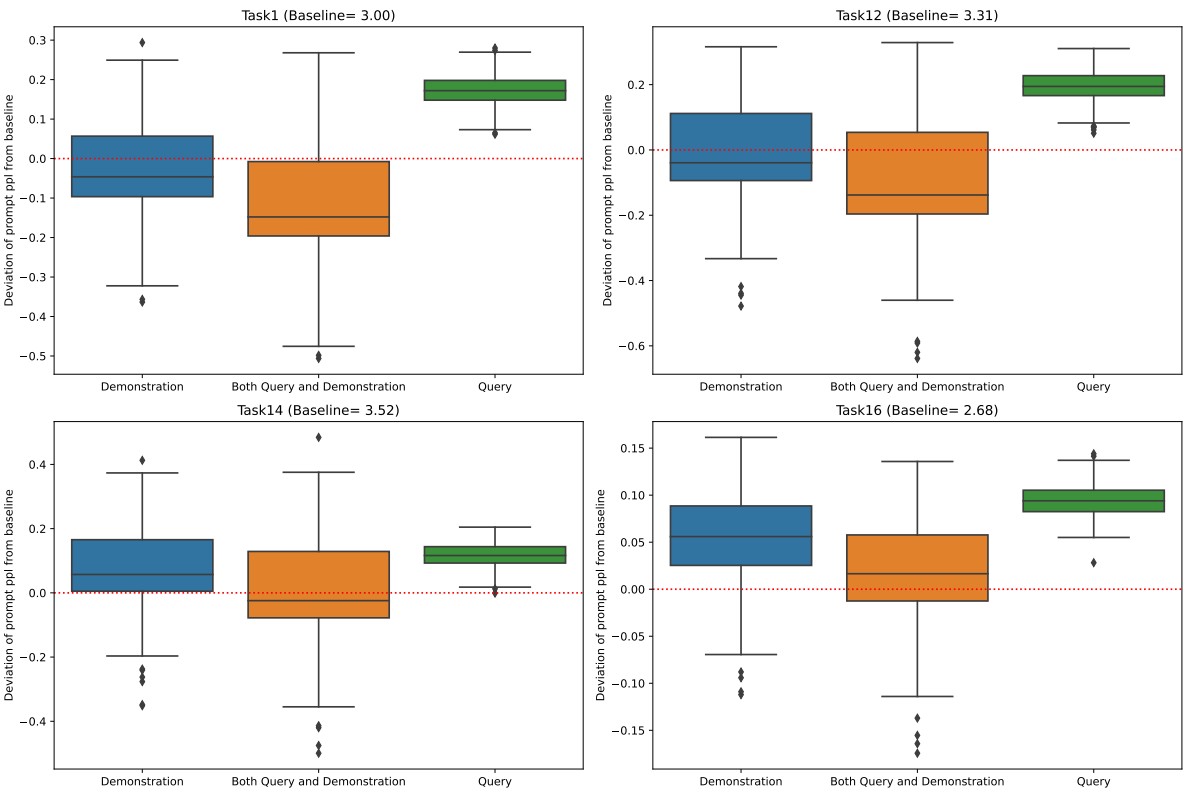

Figure 12: Deviation of prompt ppl for GPT-J-6B from the baseline across different tasks. A greater reduction in prompt ppl is observed when entities are replaced in both demonstrations and queries, a trend likely driven by the recurrent usage of the same replaced named entity within the prompt.