# OpenReview forum: "Robustness of Named-Entity Replacements for In-Context Learning"
_EMNLP/2023/Conference — EMNLP 2023 Findings_

### Official Review · Reviewer_a4iF · 2023-08-04

**Soundness:** 3

**Excitement:**

3: Ambivalent: It has merits (e.g., it reports state-of-the-art results, the idea is nice), but there are key weaknesses (e.g., it describes incremental work), and it can significantly benefit from another round of revision. However, I won't object to accepting it if my co-reviewers champion it.

**Paper Topic And Main Contributions:**

This paper explores the robustness of the in-context learning paradigm by focusing on entities, and finds that LLMs exhibit a consistent lack of robustness to entity replacements for in-context learning.

The paper conducts the experiments on three datasets with both an encoder-decoder LLM (Flan-T5) and a decoder-only LLM (GPT-J), and also tries to explain the results by analyzing frequency, token length and perplexity of the replaced entities.


**Questions For The Authors:**

Question A. Can the entity replacement in the paper be applied to other types of entities?

Question B. In Figure 2, why the entity replacement strategy D has a lower accuracy than other two strategies?


**Reasons To Accept:**

The paper explore the robustness of the in-context learning paradigm by focusing on entities, which may be a new research direction for enhancing LLM downstream performance by searching over a range of entities.

The paper conducts extensive experiments on three datasets, and did an in-depth analysis.


**Reasons To Reject:**

(1) The experiments and analysis are limited to first names, which will affect the extensibility and applicability of the methods.

(2) More discussion should be provided in Section 3, which just describes the phenomenon. For example, In Figure 2, what’s the possible reason that the entity replacement strategy D has a lower accuracy than other two strategies.


**Reproducibility:**

4: Could mostly reproduce the results, but there may be some variation because of sample variance or minor variations in their interpretation of the protocol or method.

**Reviewer Confidence:**

4: Quite sure. I tried to check the important points carefully. It's unlikely, though conceivable, that I missed something that should affect my ratings.

---

### Official Review · Reviewer_iCLV · 2023-08-10

**Typos Grammar Style And Presentation Improvements:** The authors have a good language and …
**Soundness:** 4

**Excitement:**

2: Mediocre: This paper makes marginal contributions (vs non-contemporaneous work), so I would rather not see it in the conference.

**Paper Topic And Main Contributions:**

The paper investigates how in-context learning of LLMs is impacted by entity changes. The authors contribute a sound experiment (several dataset and two sota models) with a short analysis of the results.

**Questions For The Authors:**

Do you plan on open sourcing the code to facilitate reproducibility?

**Reasons To Accept:**

-Clear aim and presentation style
-Experiments are performed on a quite large number of datasets making the results sound.
-Good efforts to account for noise in the training set-up.


**Reasons To Reject:**

 - Premise that name changes shouldn’t alter the answer is not necessarily true. There are entities and first names (Jason (Bourne/Statham) Action Heroes) that are connected to certain characteristics, which should alter the answer probability. This phenomenon is often used to write jokes or tales (e.g. the Fox, Loki, The Crow, etc. for wittiness, or Ronaldinho for his football talent, Giovanni as an Italian, Huckleberry Finn). “The Fox has the ball. The Fox went to the kitchen. Where is the ball? Probably not in the kitchen because the Fox is a known symbol for trickery.” Furthermore, there are first names like Alice and Bob, who are often used in the context of word problems in mathematical tests.
- The analysis of the problem seems superficial. Assuming the premise would be right the authors don’t account for contextual assumptions inherent in names. For example, the embedding of Johnson is probably more related to chemicals due to the company Johnson&Johnson in comparison to Wendy who is probably more related to food because of the fastfood chain. They should have grouped contextual similar first names and test for intra cluster differences. This semantic shift per first name would not necessarily show in a metric like perplexity (The results hint at correlation like this as the best performing entities vary per dataset.)
-The authors don’t indicate if they share the dataset making the impact of the paper quite small


**Reproducibility:**

3: Could reproduce the results with some difficulty. The settings of parameters are underspecified or subjectively determined; the training/evaluation data are not widely available.

**Reviewer Confidence:**

3: Pretty sure, but there's a chance I missed something. Although I have a good feel for this area in general, I did not carefully check the paper's details, e.g., the math, experimental design, or novelty.

---

### Official Review · Reviewer_4TcR · 2023-08-11

**Soundness:** 4

**Excitement:**

4: Strong: This paper deepens the understanding of some phenomenon or lowers the barriers to an existing research direction.

**Paper Topic And Main Contributions:**

This paper explores the robustness of in-context learning (ICL) for named-entity replacement. Specifically, the authors find that ICL exhibits bias towards specific entities in several entity reasoning test sets. The contributions are as follows:

1. The authors conducted exhaustive experiments, attempting to contrast the performance differences in ICL for entity replacement using two underlying models, Flan-T5-XL and GPT-J.
2. They designed reasonable constraints to ensure the semantic-preserving nature of entity replacement, such as the `Grammatical Correctness Check`.
3. The authors did not find a significant influence (correlation) of entity frequency, prompt perplexity, and token length on ICL entity replacement.

**Questions For The Authors:**

Refer to W3. Would like to see possible discussions on NE replacement on **different** parts of the exemplar.

**Reasons To Accept:**

S1. The robustness of entity replacement in in-context learning is indeed an area that urgently needs exploration. Although the authors were unable to identify the root cause, this provides a basis for future prospects in the robustness of ICL for LLMs.

S2. The writing is easy to follow with clear motivation.

S3. The authors provide detailed steps and results in the appendix, which I believe interested readers could easily reproduce and get findings.

**Reasons To Reject:**

W1. The authors failed to identify the root cause of the performance change due to entity replacement in ICL. However, it is not a major flaw, especially considering this paper is a short submission.

W2. Possibly constrained by budget, the authors did not test the robustness of the OpenAI API in terms of entity replacement. I believe that as the model and data scale, the impact of entity replacement on validation results will become increasingly minor.

W3. The comparison among `D only`, `D and q`, `q only` settings is not clear. Would like to see some possible reasons (discussions) on this, e.g. why `D only` and `q only` settings have different results on different benchmarks.

**Reproducibility:**

5: Could easily reproduce the results.

**Reviewer Confidence:**

3: Pretty sure, but there's a chance I missed something. Although I have a good feel for this area in general, I did not carefully check the paper's details, e.g., the math, experimental design, or novelty.

---

### Meta-Review · Area_Chair_5qRb · 2023-09-19

**Recommendation:** 3

**Metareview:**

The paper studies the robustness of LMs to choice of named entities (e.g. person name) and their effect showing that there is high variance in downstream performance when NEs are substituted. This is an interesting finding motivating the community to build stronger ways to validate models or build more robust models. The paper presents the findings only, but does not present any suggestion on how this could be improved. I think this is a reasonable contribution for a short paper. The paper presents other analyses regarding the frequency and token length.  There is a reasonable consensus among reviews that the the methodology is sound, but the excitement scores are mixed. I agree that the work has potential, but without mitigation, it is hard to take any action about this finding other than to check for it in other models.

Reviews iCLV raises some concerns that we do want models to be sensitive to NEs. Premise that name changes shouldn’t alter the answer is not necessarily true. I think this is an interesting discussion point that the authors should further address where name sensitivity is recommended and where models should be expected to be insensitive to this.

---

### Decision · Program_Chairs · 2023-10-07

**Decision:**

Accept-Findings

**Comment:**

The paper studies the robustness of LMs to choice of named entities (e.g. person name) and their effect showing that there is high variance in downstream performance when NEs are substituted. This is an interesting finding motivating the community to build stronger ways to validate models or build more robust models. The paper presents the findings only, but does not present any suggestion on how this could be improved. I think this is a reasonable contribution for a short paper. The paper presents other analyses regarding the frequency and token length.  There is a reasonable consensus among reviews that the the methodology is sound, but the excitement scores are mixed. I agree that the work has potential, but without mitigation, it is hard to take any action about this finding other than to check for it in other models.

Reviews iCLV raises some concerns that we do want models to be sensitive to NEs. Premise that name changes shouldn’t alter the answer is not necessarily true. I think this is an interesting discussion point that the authors should further address where name sensitivity is recommended and where models should be expected to be insensitive to this.